# Post-ERCP Pancreatitis: Prevention, Diagnosis and Management

**DOI:** 10.3390/medicina58091261

**Published:** 2022-09-12

**Authors:** Oscar Cahyadi, Nadeem Tehami, Enrique de-Madaria, Keith Siau

**Affiliations:** 1St. Josef-Hospital, A Hospital of the Ruhr-University Bochum, 44791 Bochum, Germany; 2Department of Gastroenterology, University Hospital Southampton NHS Foundation Trust, Southampton SO16 6YD, UK; 3Department of Gastroenterology, Dr. Balmis General University Hospital and Department of Clinical Medicine, Miguel Hernández University, 03010 Alicante, Spain; 4Alicante Institute for Health and Biomedical Research (ISABIAL), 03010 Alicante, Spain; 5Department of Gastroenterology, Royal Cornwall Hospitals NHS Trust, Truro TR1 3LJ, UK; 6Medical and Dental Sciences, University of Birmingham, Birmingham B4 6BN, UK

**Keywords:** post-ERCP pancreatitis, ERCP, pancreatic stenting, non-steroidal anti-inflammatory drugs

## Abstract

Endoscopic retrograde cholangiopancreatography (ERCP) carries a post-ERCP pancreatitis (PEP) rate of 2–10%, which could be as high as 30–50% in high-risk cases. PEP is severe in up to 5% of cases, with potential for life-threatening complications, including multi-organ failure, peripancreatic fluid collections, and death in up to 1% of cases. The risk of PEP is potentially predictable and may be modified with pharmacological measures and endoscopist technique. This review covers the definition, epidemiology and risk factors for PEP, with a focus on the latest evidence-based medical and endoscopic strategies to prevent and manage PEP.

## 1. Introduction

Endoscopic retrograde cholangiopancreatography (ERCP) is the primary therapeutic approach for disorders affecting the pancreatobiliary tree, including stone clearance and relief of benign and malignant biliary obstruction. Of all the mainstay endoscopic modalities, ERCP carries the highest risk of complications and mortality, with post-ERCP pancreatitis (PEP) being the most frequent complication after sedation-related adverse events, even after a seemingly straightforward procedure [1]. Although most patients with PEP take an inconspicuous or mild clinical course, some develop severe complications, such as multi-organ failure, pancreatic and/or peripancreatic fat necrosis, collections and even death. This review focuses on the diagnosis and medical management of PEP and evidence-based measures to prevent PEP.

## 2. Diagnosis of PEP

A consensus paper in 1991 defined PEP as “clinical evidence of pancreatitis” after ERCP associated with a three-fold increase of serum amylase at ≥24 h and necessitating hospital admission or prolonged hospital stay [2]. Thereafter, in 1996, Freeman added pain (i.e., new or worsening abdominal pain) as a further criterion to the PEP definition [3]. The 2020 ESGE guideline on ERCP-related adverse events defines PEP as a condition that is associated with new or worsened abdominal pain combined with elevated pancreatic enzymes (amylase or lipase ≥ 3 times upper limit of normal), thus prolonging a planned hospital admission or necessitating hospitalization after an ERCP [1].

In terms of diagnosing PEP, abdominal discomfort is common after ERCP; thus, clinical assessment, in combination with serum amylase and/or lipase, is essential to differentiate between transient post-procedural bloating; PEP; and other complications, e.g., perforation, cholangitis and unresolved biliary obstruction (such as from retained CBD stones). Early cross-sectional imaging can be helpful for diagnosis and to exclude a structural cause for PEP, e.g., retained stone, which may necessitate early repeat ERCP. The management of PEP (discussed below) is similar to that of acute pancreatitis. Endoscopists should be encouraged to clearly document the difficulty level of ERCP, type of ampulla, number of attempts required to achieve selective biliary cannulation, biliary cannulation technique, use of air vs. carbon dioxide and time required to complete the procedure, because these factors are predictors of difficult ERCP and possible PEP. 

PEP can be classified by severity. The consensus paper initially defined mild and moderate PEP solely on the duration of the hospitalization (i.e., hospital stay to 2–3 days or 4–10 days, respectively). Severe PEP was defined as hospitalization > 10 days or hemorrhagic pancreatitis or pseudocyst requiring intervention (percutaneous drainage or surgery). The revised Atlanta classification sees local complications, systemic complications and organ failure and its duration or the absence thereof at 48 h as factors to stratify the severity of acute pancreatitis [4]. Severe pancreatitis occurs in approximately 5% of PEP cases [5] and is defined by the presence of persistent (>48 h) organ failure, moderate as transient (≤48 h) organ failure or local or systemic complications and mild as the absence of complications [4]. The revised Atlanta classification appears to better predict the severity and mortality of PEP compared to the consensus criteria (Table 1) [6].

### 2.1. Pathophysiology of PEP

PEP is thought to result from an interplay of mechanical obstruction and/or hydrostatic injury, which causes early activation of pancreatic enzymes, leading to local and potentially systemic inflammation [7]. Obstruction can be caused by oedema or trauma to the papilla most often through over-manipulation. Thus, it is crucial to recognize this and to consider alternative cannulation techniques when standard attempts fail. Hydrostatic injuries can be induced by pancreatic duct (PD) injection with the use of contrast agents or water, especially in the case of acinarization. Further causes for injuries include perforation of the pancreatic duct side branch with guidewire, use of electrocautery and possibly allergic reaction to the contrast agent [8].

### 2.2. Incidence and Mortality of PEP

Cotton and colleagues (1991) analyzing the complications of biliary sphincterotomy (EST) in over 11,400 ERCP reported a PEP rate of 2.1% and a mortality rate of 0.2% [2]. Freeman and colleagues (1996) analyzing over 2300 ERCP showed a PEP rate of 5.4% with a mortality rate of <0.1% [3]. A systematic review of RCTs in 2015 with almost 13,300 patients revealed a PEP rate of 9.7% and an overall mortality rate of 0.7% with an interestingly differing PEP and mortality rate according to geographic locations with 8.4% and 0.2% in Europe, 9.9% and 0% in Asia, and 13% and 0.1% in North America, respectively [5]. Another systematic survey of prospective studies with almost 17,000 patients reported a lower PEP incidence of 3.47% [9]. A large American retrospective study comprising over 1.2 million patients between 2011 and 2017 concluded that the mortality rate increased from 2.8% of PEP patients to 4.4% at the end of study period, despite the PEP rate being 4.5% and thus comparable to previous publications. In patients with Sphincter of Oddi dysfunction (SOD), the reported PEP rate was as high as 15% [10,11]. A recent Japanese RCT with 370 patients undergoing biliary stenting revealed that patients without biliary sphincterotomy conveyed a PEP rate of 20.6% compared to 3.9% in patients with prior sphincterotomy [12]

### 2.3. Risk Factors Associated with PEP

Because of the potentially severe but modifiable nature of PEP, it is important to recognize its risk factors, most of which can be patient-related or procedure-related (Table 2).

## 3. Patient-Associated Factors

Patient-related factors include female gender, previous pancreatitis, previous PEP and suspicion of SOD, younger age, non-dilated common bile duct, normal bilirubin and end stage renal disease [1]. Other modifiable factors, such as alcohol and cocaine use, and non-modifiable factors, including race, obesity and congestive heart failure, may also be implicated [13]. A comparison of ESGE and ASGE guidelines in ERCP-related adverse events showed similar patient- and procedure-related factors [1,14]. ESGE further classifies patient- and procedural-related factors into “definite” and “likely” groups. An ERCP can be considered as high-risk for PEP if one definite factor (either patient- or procedure-related) or two likely factors are fulfilled (Table 3).

## 4. Endoscopist-Associated Factors

It is plausible that less experienced endoscopists would incur higher rates of PEP and other adverse events (AEs). Lee et al. [15] found that endoscopist with lesser experience, possibly due to difficulty in bile duct cannulation, had a higher PEP rate compared to more experienced endoscopists (OR 1.63; 95% CI, 1.05–2.53). This could partially be explained that lesser experience leads to prolonged cannulation time, which is associated with a higher PEP rate [16]. A meta-analysis by Keswani et al. found that high-annual volume endoscopists had a 60% higher ERCP success rate than low-annual-volume endoscopist and 30% less overall chance of an AE, but there was no difference in PEP when stratified by high-volume endoscopists or centers [17]. However, there was considerable heterogeneity in the definition of high-volume vs. low-volume endoscopists amongst the included studies, with cutoffs ranging from 25 to 156 annual procedures.

Several studies have explored whether trainee involvement in ERCPs may increase the risk of PEP. Using the national inpatient sample with over 480,000 ERCP in the USA, a study looked into the so-called “July effect” and post-ERCP sepsis [18]. The “July effect” is an academic period between July and September which marks the enrollment of new fellows. The study found a higher PEP and post ERCP-sepsis rate compared to a period of October–June (1.2% vs. 1.1%, *p* = 0.004 and 9.4% vs. 8.8%, *p* < 0.001, respectively). However, a multicenter study from Europe showed that ERCP success and AE was similar in both trainee and non-trainee groups [19]. This was also replicated in a recent Chinese study involving 4000 ERCPs [20]. With adequate trainer supervision and taking over the procedure as required, it is possible to maintain patient safety while delivering hands-on training until trainees are deemed to be competent for independent practice [21].

## 5. Procedure-Associated Factors

Since overmanipulating the papilla is a risk factor for PEP, it is important to study the papilla and optimize conditions before attempting cannulation. Haraldsson and colleagues (2017) proposed a visual system to classify the papilla (Figure 1). Using this classification, two recent studies indeed showed a higher rate of difficult cannulation and PEP for a protruding-type and small-type papilla [22,23], but on multivariable analysis, papilla morphology was not a significant risk for any complication [22]. An earlier German study classifying the papilla according to size and roof concluded that these had no effect on successful biliary cannulation, whereas stable scope position and visualization of the papilla were predictive [24].

Biliary cannulation can be technically challenging. ESGE defines difficult cannulation as cannulation time > 5 min, >5 contacts with the papilla or ≥1 accidental PD cannulation (the so-called “5-5-1” rule) [25]. An analysis of 1067 patients found that PEP rate was 3.9% for cannulation times between 3 and 5 min and as high as 11.9% after 5 min of cannulation attempts. PEP rate was as high as 16% in patients with > 5 min cannulation attempts and a PD cannulation [16]. Therefore, early adoption of a rescue cannulation technique (mostly precut fistulotomy) and/or change of operator should be considered according to local expertise available [15]. In trainees’ hands-on procedures, a recent study proposed a “15-10-2” rule (i.e., 15 min of cannulation attempts, 10 contacts with the papilla and ≥2 accidental PD cannulations), as rates of successful biliary cannulation, PEP and overall AE were similar to experienced endoscopists, using the “5-5-1“ rule [20].

The degree of PD manipulation directly correlates with PEP rates. In patients with a small CBD (<9 mm), PEP rates range from 4.6% without any PD manipulation to 8.3% with contrast material alone to 16.9% with guidewire alone to 22.1% with both contrast material and guidewire [26].

## 6. Prevention of PEP

### 6.1. Patient Selection

The best way of preventing PEP is by avoiding unnecessary ERCP. There should be measures in place to ensure appropriate referrals and case selection, with access to multidisciplinary team review, to ensure that ERCP is absolutely indicated for strategic planning of complex cases and to ensure that cases are appropriate for the endoscopist’s skill set [21]. Safer alternatives to ERCP, such as EUS or MRCP for confirming choledocholithiasis, should be considered if available and accessible [27]. A Japanese study reported that, in patients with a negative MRCP, EUS found stones in ~35% of the cases, and on the contrary, no stone was found in MRCP after a negative EUS [28]. Despite this, single session EUS/ERCP sessions for low-risk bile duct stones are limited by availability, expertise, reimbursement and logistical planning. Surgery may be an alternative to ERCP for CBD stones and malignant strictures. Laparoscopic CBD exploration in conjunction with cholecystectomy for the intraoperative removal of CBD stones can be performed [1]. For surgically fit patients with resectable malignant strictures, those with lower levels of bilirubin may benefit from early surgery, leaving ERCP for patients who have to wait longer until surgery or with complications e.g., cholangitis, severe pruritus or have a very high level of bilirubins prone to cholangitis and/or acute kidney failure [29]. A Dutch RCT of patients with resectable pancreatic carcinoma, which compared preoperative ERCP with fully covered biliary metal stents (FC-SEMS) vs. plastic stent (PS) vs. early surgery found that the early surgery group had an overall lower risk of AEs compared to patients assigned to FC-SEMS and PS, respectively [29].

Endoscopists should know the case in advance, study the relevant imaging, plan the team and necessary accessories and prepare a procedural roadmap. Patients requiring general anesthesia or at high ASA risk should undergo anesthetic review to ensure patient safety and comfort. Risk factors for PEP should be considered to provide an approximate estimate of risk which helps for counselling and consenting the patient and scheduling post-procedure aftercare. Performing a team timeout before the procedure could help ensure that everything needed is in place or within a short reach and align the team’s mind to reach the goal of the ERCP.

### 6.2. Medical Prophylaxis of PEP

#### 6.2.1. Non-Steroidal Anti-Inflammatory Drugs

The use of non-steroidal anti-inflammatory drugs (NSAIDs) for PEP prophylaxis has been enshrined into ERCP practice. The recent ESGE guidelines cited as many as 27 meta-analyses showing reduced PEP with NSAID prophylaxis, with an NNT to prevent PEP of 8 to 21 [1].

In addition to rectal suppositories, other routes of administration have also been studied. Oral diclofenac [30,31], celecoxib [32] or a combination of udenafil and aceclofenac [33] have not been found to reduce PEP compared to a placebo or saline infusion. Placebo-controlled studies have also found no significant difference in PEP rates with intravenous valdecoxib [34], or with the combination of intramuscular diclofenac and isotonic saline [35]. Geraci and colleagues performed a five-arm study (*n* = 20 per arm) comparing diclofenac given orally, intramuscular, intravenous, rectal and placebo and found PEP to be lowest (i.e., 0%) in the rectal diclofenac group [36].

The optimal timing of NSAID has also been debated; a Lancet RCT of 2600 patients demonstrated an overall PEP rate of 4% vs. 8% in patients receiving pre-ERCP and post-ERCP NSAIDs, respectively (OR, 0.47; 95% CI, 0.36–0.66) [37]. A sub-analysis of 346 patients from the FLUYT trial showed a PEP rate of 8% in the pre-ERCP NSAID group vs. 18% in the post-ERCP NSAID group [38]. The dose of rectal suppositories used was 100 mg for both diclofenac and indomethacin. A recent large RCT (*n* = 1037) comparing 100 mg and 200 mg post-ERCP indomethacin in high-risk patients found no difference in PEP rates [39]. Retrospective studies from Japan on the effect of low-dose rectal diclofenac (25–50 mg) have not shown this to be effective for PEP prophylaxis [40,41,42,43].

The combination of NSAID and aggressive hydration has shown a lower OR of PEP in two recent meta-analyses [44,45]. The result of the FLUYT RCT dispelled this idea; FLUYT enrolled 826 patients with moderate-to-high-risk PEP and found no difference in PEP rates between patients randomized to rectal NSAID (8%) versus the combination of rectal NSAID and aggressive fluid therapy (9%) [46].

NSAIDs are not recommended in pregnancy >30 weeks of gestation, patients with a history of allergic or pseudoallergic reaction to NSAID such as a NSAID-exacerbated respiratory syndrome or history of severe reaction such as Lyell’s Syndrome or Stevens-Johnsons-Syndrome attributed to NSAID. In these patients and their first-degree relatives NSAID should be avoided [1]. Despite this, it is worth emphasizing that AEs from a single dose NSAIDs in our clinical experience are rare.

#### 6.2.2. Intravenous Fluids

Intravenous fluids should be considered when NSAIDs are contraindicated [1]. Two meta-analyses support the role of fluid therapy for PEP prophylaxis [47,48]. Despite no significant differences in adverse events between aggressive and standard hydration in one meta-analysis [47] and in the FLUYT-trial [46], caution should be exercised in patients with significant fluid overload, e.g., congestive heart failure, decompensated cirrhosis and severe chronic kidney disease. ESGE recommends applying 3 mL kg/h during ERCP and 20 mL/kg as a bolus after ERCP, followed afterward by 3 mL/kg/h over 8 h [1]. 

#### 6.2.3. Glyceryl Trinitrate

Topical GTN appears to reduce the contractility of the sphincter of Oddi [49] and may reduce PEP. A meta-analysis of 12 RCTs found GTN to lower the overall incidence of PEP but not significantly lower the rate of moderate and severe PEP [50], with sublingual application (albeit only in 2 out of 12 RCTs) being more effective than intravenous or transdermal routes. Due to its proposed mechanism of lowering sphincter of Oddi pressure, an incremental benefit in cases of pancreatic stenting is not clear. A United States Cooperative for Outcomes Research in Endoscopy (USCORE) evaluation of pharmacologic prevention for PEP recommended the use of sublingual nitroglycerin in patients with NSAID allergy or in cases where pancreatic stenting is not possible, as well as additive prophylaxis to NSAIDs in high-risk patients, who do not receive pancreatic stenting [51]. GTN can lead to hypotension and headache; thus, it should be used with caution in these contexts, especially with intravenous and sublingual formulations [50].

#### 6.2.4. Other Agents

There are further pharmacological agents that were studied regarding PEP, such as somatostatin and octreotide (an inhibitor of exocrine pancreas function). The largest study in somatostatin to reduce PEP was a multicenter, open-label RCT with over 900 patients, which showed a 4% PEP rate in the somatostatin group vs. 7.5% in the control group [52]. In the study, somatostatin was given as a 250 μg bolus injection before ERCP and as a 250 μg/hour intravenous infusion for 11 h after ERCP [52]. A meta-analysis regarding somatostatin in PEP prophylaxis revealed an overall risk reduction with somatostatin (OR, 0.6; 95% CI, 0.41–0.89), but this was only significant in high-risk patients and not in low-risk patients [53]. Despite this, the ESGE currently does not recommend somatostatin due to the uncertainty of the estimates (upper-bound CI of the meta-analysis was close to 1) [1]. Aligning with this recommendation, a recent Iranian RCT (*n* = 376) reported no difference in rates of PEP in patients receiving intravenous somatostatin (250 ug bolus and 500 ug infusion over 2 h) and rectal indomethacin (11.4%) versus 100 mg rectal indomethacin alone (15.2%; *p* = 0.666) [54]. Protease inhibitors such as gabexate [55], ulinastatin [56] and nafamostat [57] and topical epinephrine (which reduces papillary oedema) [58] have also been studied but are not recommended by ESGE due to uncertain efficacy [1].

## 7. Procedural Factors to Prevent PEP

### 7.1. Approaches for Difficult Biliary Cannulation

The ESGE recommends wire-guided biliary cannulation because of higher success rate and avoidance of pancreatic duct contrast injection [1]. Before cannulation, it is crucial to have a stable scope position, study the papilla’s morphology, identify the orifice and plan the trajectory of cannulation to avoid excessive papillary trauma. Depending on access and ampullary morphology, it may be reasonable to start with an alternate sphincterotome or with a slimmer (0.025″) hydrophilic guidewire [59].

It is prudent to define difficult cannulation using the “5-5-1” or “15-10-2” rule in the presence of a trainee, which then should prompt second-line access strategies. In these cases, early precut-papillotomy or needle-knife fistulotomy (NKF) was shown to reduce PEP [60,61], but this requires a higher level of training and expertise. A comparison of different expertise levels in primary NKF for bulging papillae (i.e., Haraldsson Type 3) and conventional sphincterotomy has shown a higher PEP rate after conventional sphincterotomy in the low-expertise group but, interestingly, no difference in outcomes with primary NKF [62]. Repeating ERCP on another day, typically after 2–4 days, is another viable consideration. Deng [63] and Colan-Hernandez [64] reported successful biliary cannulation in approximately 80% on repeat procedures after failed index ERCP with a biliary precut papillotomy. It is recommended to stop the procedure after a prespecified time (e.g., 45 min) if cannulation does not succeed and reattempt another day.

### 7.2. Inadvertent PD Cannulation

In the case of inadvertent pancreatic duct (PD) cannulation, it is advisable to adopt an early pancreatic guidewire-assisted technique, such as double-guidewire technique (DGW) or transpancreatic biliary septotomy (TPS). For the DGW technique, the guidewire is left in the PD as a reference point, and biliary cannulation is then performed by using a second guidewire [65]. This offers several advantages, including (1) estimation of the location of the bile duct (often at the upper left side) relative to the PD orifice, (2) straightening the angle of the bile duct and (3) partially occluding the PD orifice to redirect the second wire into the CBD. A recent RCT showed that early DGW after one inadvertent PD cannulation led to higher rates (84%) of successful biliary cannulation in 10 min compared with repeated single-wire cannulation (50%) without affecting PEP rates [66]. 

TPS is a technique in which papillotomy is performed from the PD toward the direction of the bile duct. This is thought to cut the septum between the two sphincters and open the bile duct orifice, facilitating easier cannulation. In an RCT comparing TPS with DGW in difficult cannulation, higher rates of biliary cannulation were achieved with TPS (84.6% vs. 69.7%) with comparable rates of PEP (13.5% vs. 16.2%) [67]. A recent meta-analysis (comprising four RCTs with 260 patients) showed a higher successful biliary cannulation rate in the TPS vs. DGW (93% vs. 79%), with lower PEP rates (8.9% vs. 22.2%). Of note, the use of prophylactic PD stenting and/or NSAID was not clearly mentioned in the cited studies. In the RCT in which all patients received PD stenting, the risk of PEP was low (approximately 2.9%) in both groups [68]. 

In order to perform TPS or DGW successfully, the guidewire must first be secured in the PD. After completing the procedure, it is reasonable to place a pancreatic stent for PEP prophylaxis [25]. A European multicenter study comparing pancreatic stent placement vs. no stent placement after inadvertent PD cannulation showed a reduced PEP rate of 12% to 25%, although it is unclear if NSAIDs were administered [69]. Another RCT studied the combination of pharmacological prophylaxis and pancreatic stenting vs. pharmacological prophylaxis alone in over 400 high-risk patients and showed a comparable PEP rate of 14.3% vs. 15.9% [70]. In this study, all patients received pharmacological prophylaxis with 100 mg rectal indomethacin and 5 mg sublingual isosorbide dinitrate pre-ERCP. Both groups also had fluid therapy consisting of Ringer’s lactate 6 mL/kg/h during ERCP, followed by a 20 mL/kg bolus after the procedure and 3 mL/kg for 8 additional hours. Thus, it is unclear if PD stenting would provide incremental benefit in addition to extensive pharmacological and fluid prophylaxis.

### 7.3. Prophylactic PD Stenting

PD stents are currently the mainstay prophylactic measure for high-risk patients and nearly eliminates the risk of severe PEP [1]. In the study by Philip et al. [69], the number needed to treat (NNT) for PD stenting to prevent one case of PEP was 8.1. Despite their efficacy, variation in practice exists. A 2012 survey found that only half of UK endoscopists had ever considered prophylactic PD stenting [71]. Retained PD stents must be extracted within 2 weeks, as stent retention beyond this can result in stent-induced PD fibrosis and a 5.2-fold higher risk of PEP [72,73]. The ESGE recommends assessing for spontaneous migration of the PD stent by 5–10 days post-ERCP (with plain abdominal radiography) and for urgent extraction if retained. In the RCT by Chahal et al. [73], the placement of an unflanged short (3 cm) 5 Fr stent was found to be easier to deploy than a longer (8 cm+) 3 Fr stent and led to higher rates of stent dislodgement at 14 days (98% vs. 88%, *p* < 0.001), thus reducing the need for endoscopic extraction. The advent of biodegradable PD stents allows for rapidly degrading stents (within 12 days), thus obviating the need for abdominal radiography +/− stent extraction [74].

In cases where the guidewire did not inadvertently cannulate the PD, repeated attempts at PD cannulation solely for the purpose of PD stent placement is not advisable, as failed attempts at PD stenting can lead to an extremely high risk of PEP (up to 65%) [75]. 

### 7.4. Other Intraprocedural Modifiers

In patients requiring endoscopic papillary balloon dilation (EPBD, aka balloon sphincteroplasty), the duration of dilation appears relevant. An RCT showed that, in EPBD with a 10 mm balloon, dilation of <1 min was associated with a higher rate of PEP (15%) compared to 5 min with a PEP of 4.8% and with higher success of stone extraction in the 5 min group [76]. Another study found that dilation of <3 min had an increased PEP rate of 13% vs. 3% in the 3–5 min group [77]. A large RCT combining small sphincterotomy (3–5 mm), followed by balloon dilation with four different durations (0 s, 30 s, 60 s, 180 s and 300 s) found that 30 s balloon dilatation time after sphincterotomy had lower PEP incidence than the 300 s group (7% vs. 15%) [78]. Thus, in patients needing a combination of EPBD and EST a dilation duration of 30 s could lead to less PEP.

For biliary strictures, self-expandable metallic stents (SEMSs) may be deployed without sphincterotomy, especially in patients who are at a high bleeding risk. From retrospective studies, the rates of PEP appear to be higher with SEMS (8.0%) vs. plastic stents (4.8%) [79] but similar for covered vs. uncovered SEMS (6.9% vs. 7.5%, *p* = 0.82) [80]. In a recent RCT, sphincterotomy before stent deployment was associated with lower rates of PEP (3.9%) versus those without sphincterotomy (20.6%, *p* < 0.001) [12]. 

## 8. Management of PEP

The therapy for PEP is similar to that of acute pancreatitis. Analgesia and supportive care with fluids are often sufficient in most patients with PEP [81]. PEP severity should be assessed according to the revised Atlanta classification to identify patients with moderate or severe cases and channel them to the appropriate level of care, e.g., high-dependency or intensive care unit with organ support where necessary. Because of the criteria currently applied (local complications or persistent organ dysfunction > 48 h), classifying PEP’s severity correctly could only be performed in retrospect. Nonetheless, early identification of predicted severe pancreatitis is theoretically lifesaving. A plethora of other scores for estimating severity, e.g., APACHE-II and Ranson Score or Pancreatitis Outcome Prediction score [82], can also be used to predict severe PEP.

Fluid therapy should be started after the diagnosis is confirmed. Some evidence points toward a benefit for Lactated Ringer Solution instead of normal saline, but this is controversial [83]. An analysis of three RCTs and five retrospective studies found that, on the first day, a starting infusion rate of >300 mL/h or <200 mL/h could be harmful to the patients and recommend an infusion rate of 200–300 mL/h, which means a total volume of ca. 4800–7200 mL of fluid on the first day [84]. A multicenter trial (Waterfall trial-NCT04381169) studying fluid therapy with Ringer’s lactate compared aggressive vs. moderate fluid therapy in acute pancreatitis. The aggressive arm received 20 mL/kg bolus—administered over 2 h, followed by 3 mL/kg/h for 12 h—vs. a moderate arm receiving a bolus 10 mL/kg—administered over 2 h in case of hypovolemia or no bolus in patients with normovolemia, followed by 1.5 mL/kg/h for 12 h; afterward, all patients with normovolemia received 1.5 mL/kg/h. The final results of this trial will be available in the following weeks (accepted, under press embargo) [85].

## 9. Duty of Candor

The United Kingdom General Medical Council defines candor as openness and honesty when things go wrong. In its duty-of-candor guidelines [86], it states that every healthcare professional must be open and honest with patients in their care when something goes wrong or if it causes, or has the potential to cause, harm or distress. It stresses that it is always right to say “sorry” and gives information about things that perhaps went wrong and that this is not an admission of liability. A lot of factors leading to PEP are sometimes outside of the hand of the endoscopist and are not due to error, but the endoscopist still holds a major active role in the process. Thus, in the advent of PEP, saying sorry to the patient and his/her family and explaining the situation and the treatment of PEP with an outlook of what to come should always be considered, along with any shared learning.

## 10. Summary

PEP is a potentially life-threatening complication of ERCP which can be mitigated through a combination of pharmacological and intraprocedural measures, prompt diagnosis and early management. Efforts to reduce PEP risk has led to the publication of a plethora of high-quality RCTs in recent years, along with the release of international guidelines on PEP (Table 4). Implementation of evidence-based best practices, quality assurance and ERCP training will help to further minimize PEP risk and improve patient safety in ERCP.

## Figures and Tables

**Figure 1 medicina-58-01261-f001:**
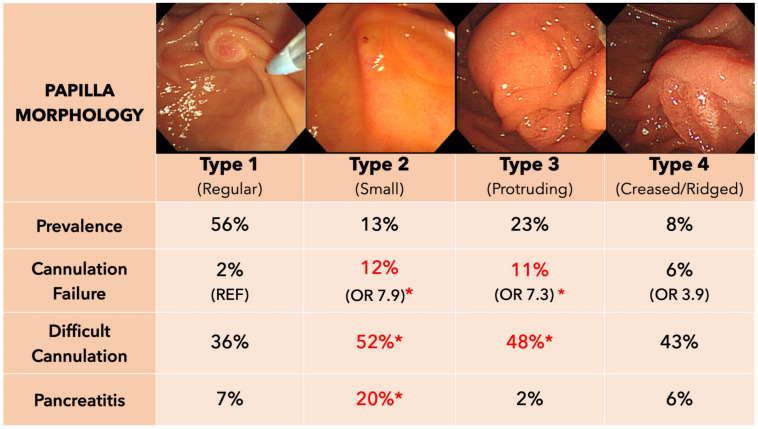
A proposed classification of the papilla morphology and the associated difficulty during bile duct cannulation. * Significantly higher risk vs. Type 1 papillae.

**Table 1 medicina-58-01261-t001:** Comparison of severity grade according to the consensus paper and the Revised Atlanta Classification.

Severity	Consensus Paper	Revised Atlanta Classification
Mild	Hospital stay up to 2–3 days	No organ failure No systemic or local complication
Moderate	Hospital stay up to 4–10 days	Organ failure * that resolves within 48 h (transient organ failure) and/or Local or systemic complications without persistent organ failure
Severe	Hospitalization > 10 days or necrotizing pancreatitis or pseudocyst or intervention (percutaneous drainage or surgery)	Persistent organ failure * > 48 h Single organ failure Multiple organ failure

* Organ failure based on modified Marshall score defined as any of the following: PaO2/FiO2 < 300, systolic blood pressure < 90 mmHg despite fluid resuscitation, serum creatinine > 170 µmol/L (>1.9 mg/dL).

**Table 2 medicina-58-01261-t002:** Excerpt of patient and ERCP-related risk factors for PEP (adapted from Dumonceau 2020 and * Mutneja 2020). OR: Odds Ratio; PEP: post-ERCP pancreatitis; SOD: Sphincter of Oddi Dysfunction.

Patient-Related Factors	OR	Procedure-Related Factor	OR
Previous history of PEP	3.2–8.7	Difficult cannulation	1.7–15
Non dilated common bile duct	3.8	Multiple pancreatic duct cannulation	2.1–2.7
Female gender	1.4–2.2	Pancreatic injection	1.6–2.7
Previous history of pancreatitis	2.0–2.90	Biliary balloon dilatation on an intact biliary sphincter	4.5
Suspicion of SOD	2.04–4.4	Failure to clear bile duct stones	4.5
Younger age	1.6–2.9	Precut Papillotomy	2.1–3.1
Black race	1.1 *	Pancreatic sphincterotomy	1.2–3.1
Obesity	1.1 *	Intraductal ultrasound	2.4
Congestive heart failure	1.3 *		
End stage renal disease	1.9 *		
Cocaine use	1.5 *		
Alcohol use	1.1 *		

**Table 3 medicina-58-01261-t003:** Definite and likely procedure- and patient-related factors for post-ERCP pancreatitis [1]. PD: pancreatic duct. PEP: post-ERCP pancreatitis. SOD: Sphincter of Oddi Dysfunction.

	Patient-Related	Procedure-Related
Definite risk factors	Suspicion of SOD	Difficult cannulation
	Previous PEP	>1 PD cannulation
	Previous pancreatitis	Pancreatic injection
	Female sex	
Likely risk factors	Younger age	Failure to complete stone clearance
	Non-dilated biliary duct	Biliary balloon dilatation of the native papilla
	Absence of chronic pancreatitis	Precut or pancreatic sphincterotomy
	Normal serum bilirubin	Intraductal ultrasound
	End stage renal disease	

**Table 4 medicina-58-01261-t004:** Summary of international guidelines on PEP prophylaxis.

	ESGE 2020 [1]	ASGE 2017 [14]	Japan 2015 [87]
Rectal NSAID	✓ before ERCP	✓	✓
Others	If contraindication to fluid and NSAID, 5 mg sublingual glycerin trinitrate before ERCP	Not stated	Not recommended
IV fluids	If contraindication to NSAID, Lactated Ringer’s During procedure: 3 mL/kg/h After procedure: 20 mL/kg Bolus followed by 3 mL/kg/h for 8 h	Periprocedural intravenous hydration with Lactated Ringer’s recommended	Not stated
Combination	Routine combination of rectal NSAID with other measures not recommended	NSAID and pancreatic stent is probably not superior to either technique alone	Not stated
Pancreatic stent	In patients with inadvertent PD cannulation or opacification and in DGW	In high-risk patients e.g., multiple PD cannulations	Only in high-risk group for PEP: Confirmed/suspicion of SOD, Difficult cannulation, Precut sphincterotomy, Balloon dilatation.
Risk stratification	Presence of at least one definite patient- or procedure-related risk factor or presence of at least two likely patient- or procedure-related risk factors For a list of the procedure- and patient-related procedures, see Table 3	A list of procedure- and patient-related risk factors is available No exact stratification is stated	No other stratification but for the group, which should receive pancreatic stent
Guidewire method	**✓**	**✓**	**✓**

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
