# Peer review of "Post-ERCP Pancreatitis: Prevention, Diagnosis and Management"

_medicina, 2022, doi:10.3390/medicina58091261_

Round 1

Reviewer 1 Report

Nice review article. Can't say the information is necessary novel, but is summarized well. PEP remains a topic of great clinical and research interest globally. No factual corrections since this is a review. Please include a summary table of rates of PEP with proposed pharmacological/endoscopic interventions.

Additionally, I have a few minor grammatical corrections =

Abstract = "which" may be as high as 30-50% in high risk cases. PEP is severe in up to 5% "cases", with potential for severe complications including multi-organ failure, peripancreatic fluid collections, and death in up to 1% "cases".

Introduction = Of all the mainstay endoscopic modalities, ERCP carries the highest risk of complications and mortality, with post-ERCP pancreatitis (PEP) being the most frequent complication, even after a seemingly straight-forward procedure (insert reference)

Definition of PEP = Severe PEP was defined as hospitalisation >10 days or hemorrhagic pancreatitis or pseudocyst or "requiring" intervention (percutaneous drainage or surgery).

Endoscopist associated factors = would incur rates of PEP and other "adverse events (AEs). Lee et al [15] found that "endoscopists with lesser experience, possibly aided by difficulty in bile duct cannulation, had a higher PEP rate compared to more experienced endoscopists (OR 1.63, 95% CI 1.05-2.53) [16]"

Doesn't need reference [17] at two separate places.

Procedure associated factors = An earlier German study classifying the papilla according to size and roof concluded that these had no effect on successful biliary cannulation; whereas stable scope position and visualisation of the pa-pilla were predictive [24]. - Correct sentence, check grammar.

Prevention of PEP= A Japanese study reported "that in patients with a negative MRCP, EUS found stones in ~35% of cases. On the contrary, no stones were found on MRCP after a negative EUS [28]"

Surgery may be an alternative to ERCP for CBD stones and ma-lignant strictures. Laparoscopic CBD exploration can be performed in conjunction with cholecystectomy for the intraoperative removal of CBD stones (Insert reference)

For surgically-fit patients with resectable malignant strictures, those with low levels of bilirubin (mostly <10 g/dl or ~171.04 μmol/l) may benefit from early surgery, leaving ERCP for patients who have to wait longer until surgery or with complications, e.g. cholangitis or have a high level of bilirubins prone to cholangitis and/or acute kidney injury. (insert reference)

Risk factors for PEP should be considered to provide an approx-imate estimate of risk which helps for counselling and consenting the patient and sched-uled post-procedure aftercare. = correct grammar

NSAIDs = NSAIDs are typically contraindicated in pregnancy, history of severe reaction, asthma and renal failure (insert reference)

Management of PEP = A multicenter trial (Waterfall trial - NCT04381169) studying fluid therapy with Ringer’s lactate compared aggressive vs. moderate fluid ther-apy in acute pancreatitis. The aggressive arm received 20-mL/kg bolus - administered over 2h, followed by 3 mL/kg/h for 12h vs. a moderate arm receiving a bolus 10 mL/kg - ad-ministered over 2h in case of hypovolemia or no bolus in patients with normovolemia, followed by 1.5 mL/kg/h for 12h; afterwards all patients with normovolemia received 1.5 ml/kg/h. The final results of this trial will be available in the following weeks (accepted, under press embargo) [88]. = recommend exclusion as results of this trial have not officially been published yet

Author Response

Nice review article. Can't say the information is necessary novel, but is summarized well. PEP remains a topic of great clinical and research interest globally. No factual corrections since this is a review. Please include a summary table of rates of PEP with proposed pharmacological/endoscopic interventions.

Response: Thank you for taking the time to review our review article. We sincerely appreciate your comments and suggestions which we feel have improved the quality of our article.

Additionally, I have a few minor grammatical corrections =

Abstract = "which" may be as high as 30-50% in high risk cases. PEP is severe in up to 5% "cases", with potential for severe complications including multi-organ failure, peripancreatic fluid collections, and death in up to 1% "cases".

Response: We have incorporated your suggestions into our abstract.

Introduction = Of all the mainstay endoscopic modalities, ERCP carries the highest risk of complications and mortality, with post-ERCP pancreatitis (PEP) being the most frequent complication, even after a seemingly straight-forward procedure (insert reference)

Response: We have incorporated the reference into our abstract.

Definition of PEP = Severe PEP was defined as hospitalisation >10 days or hemorrhagic pancreatitis or pseudocyst or "requiring" intervention (percutaneous drainage or surgery).

Response: Thank you for the suggestion. The sentence was corrected.

Endoscopist associated factors = would incur rates of PEP and other "adverse events (AEs). Lee et al [15] found that "endoscopists with lesser experience, possibly aided by difficulty in bile duct cannulation, had a higher PEP rate compared to more experienced endoscopists (OR 1.63, 95% CI 1.05-2.53) [16]"

Doesn't need reference [17] at two separate places.

Response: We have deleted one of the two references

Procedure associated factors = An earlier German study classifying the papilla according to size and roof concluded that these had no effect on successful biliary cannulation; whereas stable scope position and visualisation of the pa-pilla were predictive [24]. - Correct sentence, check grammar.

Response: We have checked the grammar.

Prevention of PEP= A Japanese study reported "that in patients with a negative MRCP, EUS found stones in ~35% of cases. On the contrary, no stones were found on MRCP after a negative EUS [28]"

Response: We have incorporated the suggestion.

Surgery may be an alternative to ERCP for CBD stones and ma-lignant strictures. Laparoscopic CBD exploration can be performed in conjunction with cholecystectomy for the intraoperative removal of CBD stones (Insert reference)

For surgically-fit patients with resectable malignant strictures, those with low levels of bilirubin (mostly <10 g/dl or ~171.04 μmol/l) may benefit from early surgery, leaving ERCP for patients who have to wait longer until surgery or with complications, e.g. cholangitis or have a high level of bilirubins prone to cholangitis and/or acute kidney injury. (insert reference)

Response: thank you for your comment. we have adapted the paragraph and inserted the reference

Risk factors for PEP should be considered to provide an approx-imate estimate of risk which helps for counselling and consenting the patient and sched-uled post-procedure aftercare. = correct grammar

Response: thank you for your comment. we have adapted the sentence 

NSAIDs = NSAIDs are typically contraindicated in pregnancy, history of severe reaction, asthma and renal failure (insert reference)

Response: Thank you. The paragraph is paraphrased and a reference was incorporated

Management of PEP = A multicenter trial (Waterfall trial - NCT04381169) studying fluid therapy with Ringer’s lactate compared aggressive vs. moderate fluid ther-apy in acute pancreatitis. The aggressive arm received 20-mL/kg bolus - administered over 2h, followed by 3 mL/kg/h for 12h vs. a moderate arm receiving a bolus 10 mL/kg - ad-ministered over 2h in case of hypovolemia or no bolus in patients with normovolemia, followed by 1.5 mL/kg/h for 12h; afterwards all patients with normovolemia received 1.5 ml/kg/h. The final results of this trial will be available in the following weeks (accepted, under press embargo) [88]. = recommend exclusion as results of this trial have not officially been published yet

Response: our current update offers an updated literature review available on PEP. Waterfall trial could be the next landmark paper on the treatment of PEP and hence citing this study would offer the readers first insight into this important study potentially positively effecting their patient treatment

Reviewer 2 Report

This is a great review summarizing current evidence in this field. I am sure this review will interest the broad audience. However, I suggest some revision be made to provide more information through this review.

1. A paragraph descrbing the difference between PEP and AP caused by other factors should be added to help the readers understand the uniqueness of PEP. 

2. In Table2, OR should be reported uniformly. I would suggest OR(95CI) for all the reported ORs.

3. Page four, it should be "....less experienced endoscopists would incur increased rates of PEP..."

4. Figure legends are missing for Figure1.

5. Page 7, a full name for GTN should be provided for a short title.

6. For the managment of PEP, similarly, the authors should focus on the uniqueness of PEP rather than common treatment strategy for AP. Some speical examinations, difference in time preseting to hosptal, etc,. should be discussed. 

Author Response

This is a great review summarizing current evidence in this field. I am sure this review will interest the broad audience. However, I suggest some revision be made to provide more information through this review.

Thank you for taking the time to review our paper. We are flattered by your kind comment and suggestions for improvement. 

1. A paragraph descrbing the difference between PEP and AP caused by other factors should be added to help the readers understand the uniqueness of PEP. 

Response: Thank you for your comments. We have added this paragraph

  1. In Table2, OR should be reported uniformly. I would suggest OR(95CI) for all the reported ORs.

Response: the table was a summary of ORs reported by different publications and hence a 95-CI is not intended

  1. Page four, it should be "....less experienced endoscopists would incur increased rates of PEP..."

Response: thank you for your comment. The sentenced was corrected

4. Figure legends are missing for Figure1.

Response: thank you for your comment. The legend was added

  1. Page 7, a full name for GTN should be provided for a short title.

Response: thank you for your comment. This is now provided in the text

  1. For the managment of PEP, similarly, the authors should focus on the uniqueness of PEP rather than common treatment strategy for AP. Some speical examinations, difference in time preseting to hosptal, etc,. should be discussed.

Response: thank you for your comment. This suggestion is incorporated in the text under management of pep

Reviewer 3 Report

This manuscript is clear and very analytical full of details. The endoscopic tecnique is widely dealwith a very precise review of the literature. I suggest to develop in a more detail the medical  management of PEP specially in the first phase , included also the role of  instrumental imaging

Author Response

This manuscript is clear and very analytical full of details. The endoscopic tecnique is widely dealwith a very precise review of the literature. I suggest to develop in a more detail the medical management of PEP specially in the first phase , included also the role of instrumental imaging

Response: Thank you for taking the time to review our paper and for your kind comments. We have incorporated this in our manuscript.